# Study on the Localization Technology for Giant Salamanders Using Passive UHF RFID and Incomplete D-Tr Measurement Data

**DOI:** 10.3390/s26010106

**Published:** 2025-12-23

**Authors:** Nanqing Sun, Didi Lu, Xinyao Yang, Hang Gao, Junyi Chen

**Affiliations:** College of Intelligent Systems Science and Engineering, Hubei Minzu University, Enshi 445600, China; 202430357@hbmzu.edu.cn (N.S.); 202230277@hbmzu.edu.cn (D.L.); 202430323@hbmzu.edu.cn (X.Y.); 202430300@hbmzu.edu.cn (H.G.)

**Keywords:** Chinese giant salamander, passive UHF RFID technology, 3D LANDMARC method, D-Tr framework, ecological observation

## Abstract

To enhance the monitoring and conservation efforts for China’s Class II endangered species, specifically the wild giant salamander and its ecosystems, this study addresses the urgent need to counteract the rapid decline of its wild population caused by habitat loss and insufficient surveillance. We present an innovative localization system based on passive Ultra-High-Frequency Radio Frequency Identification (UHF RFID) technology, employing a Double-Transform (D-Tr) methodology that integrates an enhanced 3D LANDMARC algorithm with GAIN generative adversarial networks. This system effectively reconstructs missing Received Signal Strength Indicator (RSSI) data due to environmental barriers by applying a log-distance path loss model. The D-Tr framework simultaneously generates RSSI sequences alongside their first-order differential characteristics, allowing for a comprehensive analysis of spatiotemporal signal relationships. Field tests conducted in the Hubei Xianfeng Zhongjian River Giant Salamander National Nature Reserve reveal that the positioning error consistently remains within 10 cm, with average accuracy improvements of 20.075%, 15.331%, and 12.925% along the X, Y, and Z axes, respectively, compared to traditional time-series models such as long short-term memory (LSTM) and gated recurrent unit (GRU). This system, designed to investigate the behavioral patterns and movement paths of farmed giant salamanders, achieves centimeter-level tracking of their cave-dwelling activities. It provides essential technical support for quantitatively assessing their daily activity patterns, habitat choices, and population trends, thereby promoting a shift from passive oversight to proactive monitoring in the conservation of endangered species.

## 1. Introduction

The wild giant salamander, recognized as the largest living amphibian species on the planet, is classified as a second-grade endangered species under national protection. This amphibian is predominantly found in freshwater habitats throughout China and plays a crucial role in the ecosystem [1,2,3]. The Hubei Xianfeng Zhongjian River Giant Salamander National Nature Reserve, located between Gaoleshan Town and Zhongbao Town in Xianfeng County, prioritizes the conservation of this species. Unfortunately, its population in the wild is rapidly declining due to habitat loss, pollution, and excessive harvesting driven by human activities [4,5]. Currently, the management of the reserve relies primarily on manual conservation efforts, which are hindered by a lack of foundational data on natural resources and inadequate information systems, limiting the effectiveness of conservation strategies. Ecologically, the Chinese giant salamander prefers clear, well-oxygenated mountain streams and often utilizes underwater caves or rock formations for shelter. It is primarily nocturnal, favors fast-flowing river areas, and feeds on fish, shrimp, and insects [6,7]. Due to its distinct behavioral patterns, current monitoring techniques encounter challenges in providing effective oversight, with most reports being descriptive and delayed. To date, few studies have employed positioning technology to comprehensively analyze and visualize the behavior of this species.

Current agricultural positioning technologies encompass Bluetooth, Wireless Fidelity (WiFi), Ultra-Wideband (UWB), and Ultra-High-Frequency Radio Frequency Identification (UHF RFID). Researchers have proposed an economical solution for low-accuracy cow tracking that utilizes signals from an acceleration measurement system through the Bluetooth Low Energy protocol [8]. Additionally, a remote operation system for fruit-picking robots has been developed, leveraging big data and WiFi to facilitate the positioning and management of harvesting robots in orchards via a computer interface [9]. Other studies have introduced a positioning algorithm that integrates monocular vision with UWB technology for mobile platforms in seedling greenhouses [10]. This approach involves creating visual beacons and implementing a monocular vision ranging function by calculating the pixel distance between two points on the beacon and the actual distance to the camera. The federated Kalman filtering algorithm is then employed to merge monocular vision with UWB, forming a positioning system for mobile platforms in greenhouses. Although these positioning techniques are effective for their intended applications, they encounter numerous challenges when applied to tracking giant salamanders, such as high implementation complexity, substantial signal interference from intricate field conditions, limited underwater detection capabilities, and difficulties associated with the nocturnal habits of the salamanders.

UHF RFID positioning technology offers advantages such as compact size, affordability, energy efficiency, and an extensive communication range, enabling it to address certain challenges effectively. However, traditional UHF RFID positioning encounters significant obstacles, primarily due to its dependence on numerous reference tags, which escalates costs and renders the system more vulnerable to electromagnetic interference. Additionally, outdoor barriers can lead to data loss, and the neglect of temporal aspects of Received Signal Strength Indication (RSSI) results in the omission of critical information [11,12,13].

This study introduces an innovative tracking system for giant salamanders that utilizes passive UHF RFID technology based on the Double-Transform (D-Tr) model. The system was implemented in the Zhongjian River Giant Salamander National Nature Reserve, located in Xianfeng County, Hubei Province, to monitor the behavioral patterns and living habits of giant salamanders while also gathering image and video data. The accuracy of the system in positioning and its adaptability were assessed through both laboratory tests and field trials. Future integration with target recognition technology [14] could enhance technical support for amphibian conservation research. This research was conducted with farmed giant salamanders, henceforth referred to as giant salamanders, in compliance with animal protection regulations. The key contributions of this research include the following:1.A novel passive UHF RFID positioning system specifically designed and implemented for giant salamanders is developed, tailored to intricate aquatic environments. This innovative approach facilitates detailed observation of the behavior of wild giant salamanders.2.To address the inherent challenges of signal blockage and data loss during field detection, we propose a novel approach that employs generative adversarial imputation networks (GAINs). This method effectively generates and imputes missing Received Signal Strength Indicator (RSSI) data, thereby significantly enhancing the reliability of the data for subsequent localization techniques.3.The D-Tr model is innovatively applied in the field of animal studies, enhancing its efficacy in analyzing signals and inferring locations within dynamic and non-static environments. This is achieved by simultaneously extracting spatiotemporal characteristics from the original Received Signal Strength Indicator (RSSI) data along with their first-order derivatives.4.This paper proposes a positioning structure that integrates data enhancement with advanced feature learning, collaborating with GAIN and D-Tr models to improve the 3D LANDMARC system. This approach establishes a comprehensive process that encompasses data preparation and culminates in precise positioning.5.This research seeks to create a methodological basis for the centimeter-scale, continuous, and non-invasive monitoring of endangered aquatic species, enabling conservation biology to evolve from broad qualitative assessments to detailed quantitative evaluations.

## 2. Materials and Methods

### 2.1. Design of RFID Giant Salamander Positioning System

#### 2.1.1. System Architecture

The positioning system for giant salamanders employs a decentralized framework that primarily consists of RFID tags, readers, antennas, and a central host computer. RFID tags are affixed to the tails of the salamanders for individual identification. Four antennas are strategically positioned to maximize monitoring coverage, located along the riverbanks and within the water of the reserve, ensuring effective communication with the tags [15,16]. The readers collect information from the tags and can measure the distance between the tags and antennas [17,18]. Data gathered by the antennas are transmitted to the host computer via Ethernet cables, where the Intelligent Monitoring System for the Giant Salamander Reserve processes, stores, and analyzes the information. This configuration enables real-time tracking of salamander locations, monitoring of their movements, and the generation of data reports. The process of positioning giant salamanders using RFID technology is illustrated in Figure 1.

#### 2.1.2. Selection of RFID Tags

Considering the habitat and biological characteristics of the Chinese giant salamander, this research selected the AP-NL01 model of UHF RFID tag, which is recognized for its compact size, waterproof design, and strong resistance to interference. To accommodate the breeding conditions of the salamander, the ultra-high-frequency range was chosen to minimize the impact of water on the radio frequency signal strength. Both the tag and the accompanying cable tie are made from TPU material, which is biodegradable and exhibits excellent compatibility with biological systems, ensuring no harm or discomfort to the salamander [19,20]. The actual RFID tag is illustrated in Figure 2.

#### 2.1.3. Selection of Readers and Antennas

Given that higher precision and a larger detection range are more favorable for the antenna readings of giant salamanders, a planar antenna was selected for monitoring purposes. The design of the antenna and its physical counterpart are illustrated in Figure 3 below. This study utilizes the IE704 model four-channel reader. The relevant parameters of both the antenna and the reader are presented in Table 1 and Table 2, respectively.

## 3. Construction of the Giant Salamander Positioning Algorithm

### 3.1. Principle of 3D LANDMARC Positioning Basic Algorithm

RFID positioning typically employs the 3D LANDMARC algorithm, which integrates both range-based and range-free techniques for identifying RFID tags. This system involves placing reference tags at predetermined locations and analyzing the real-time variations in the Received Signal Strength Indicator (RSSI) between the target and reference tags. It utilizes the k-nearest neighbor (kNN) algorithm to identify the nearest reference points in a three-dimensional context, ultimately determining the 3D coordinates of the target tag [21,22,23]. While this method offers adequate accuracy for standard positioning needs, it may fall short for applications that demand higher precision. Additionally, outdoor obstacles can impede antennas from receiving tag information, leading to potential data loss.

### 3.2. Improved 3D LANDMARC Positioning Algorithm

This study presents the GAIN generative adversarial network as a solution to data loss, utilizing the log-distance path loss model to accurately estimate missing data, thereby improving the reliability and utility of the collected information. Furthermore, to address outdoor positioning challenges posed by environmental factors, the research employs the D-Tr model, which is particularly effective in managing time-series data and provides a significant improvement in positioning accuracy compared to conventional techniques [24,25,26].

#### 3.2.1. Introduction of GAIN Generative Adversarial Network

In intricate outdoor settings, signal transmission often encounters interference from obstacles such as tree limbs and submerged barriers, complicating antennas’ ability to achieve consistent signal reception. This challenge can lead to the loss of monitoring information, adversely affecting the training of future dual-stage transformer models. To address this issue, enhancing the input data is crucial. To improve the quality of the generated data, this study employs the GAIN method, an advanced approach derived from generative adversarial networks (GANs). The structure of the GAIN is illustrated in Figure 4.

The log-distance path loss model is as follows:(1)$$\mathrm{PL}(d)\;=\;\mathrm{PL}(d_{0})\;-\;10n\mathrm{lg}\frac{d}{d_{0}}\;-\;N_{0}$$

In this context, PL(d) indicates the signal power measured at a distance of *d* meters from the antenna, expressed in dBm. Similarly, $\mathrm{PL}(d_{0})$ at a distance of *d*_0_ m from the antenna is also represented in dBm. The parameter *n* refers to the exponent of signal loss, while *d* and *d*_0_ represent the respective distances from the antenna in meters. Additionally, *N*_0_ signifies the effect of shadow fading.

Upon entering the initial matrix, it is divided into three distinct matrices: the data matrix, the random matrix, and the mask matrix. The mask matrix uses 0 s to indicate absent data and 1 s to denote present data, which assists in identifying missing information. The data matrix retains known values while replacing unknown elements with 0 s. In contrast, the random matrix substitutes missing values with random numbers and converts known data into 0 s. These three matrices are subsequently input into a generator to produce a completed matrix. Data are generated using the following formulas:(2)$$\bar{X}\;=\;G\left(X,M,\left(1\;-\;M\right)⊙Z\right)$$(3)$$\hat{X}=M⊙X+\left(1-M\right)⊙\bar{X}$$

In this context, *G* represents the generator, *M* stands for the mask matrix, *X* refers to the data matrix, and *Z* indicates the random matrix, while element-wise multiplication is denoted by ⊙. Additionally, the vector of estimated values is represented by $\bar{X}$, and the complete data vector is indicated by $\hat{X}$.

Upon processing three matrices, the generator produces an imputed matrix along with a hint matrix. These matrices are then utilized by the discriminator to evaluate the validity of the imputed matrix. The hint matrix is derived from the mask matrix, which serves to prevent the generator from directly overpowering the discriminator. This approach enhances adversarial iterations, limits the generator’s ability to produce realistic data, and improves the data generation and discrimination functions of the GAIN.

#### 3.2.2. Introduction of Double-Transform Model

This study collects data on the Received Signal Strength Indicator (RSSI) measurements of reference tags, their corresponding distances, predetermined distance intervals, and the RSSI values associated with those intervals. It analyzes these characteristics and utilizes them to train time-series deep neural networks for predictive modeling [27,28,29].

In the realm of time-series analysis, first-order differentiation involves calculating the differences between successive elements in a discrete series. This measure is crucial for analyzing information, as it accurately reflects the relationships between data points, thereby enhancing the stability of the sequence. To address the erratic variations in Received Signal Strength Indicator (RSSI) values, we employ a comprehensive strategy that incorporates first-order differential data. We introduce an innovative Dual-stage Transformer (D-Tr) architecture, as illustrated in Figure 5, to improve the analysis and processing of these signal features.

The fundamental concept of the dual-stage Transformer positioning framework is predicated on two concurrent time-series analysis components: the original Transformer module and the first-order Transformer module. These components are meticulously designed to capture distinct dynamic characteristics of signals and integrate their features for position estimation. The original module processes the raw signal strength sequence [RSSI(t − T), RSSI(…), RSSI(t)] over a defined time window T, generating features [H(t − 1), H(t), H(…), H(T)] that reflect the signal’s state during that interval. Meanwhile, the first-order module analyzes the sequence of signal strength differences [ΔRSSI(t − T), ΔRSSI(…), ΔRSSI(t)] over the same duration, producing a feature ΔH(T) that indicates the signal’s dynamic change trend. The model then merges the high-dimensional time features H(T) and ΔH(T) from these two separate modules, each with a dimension of T × d, to create a unified feature representation. This combined feature is subsequently input into a fully connected network layer, which ultimately generates the predicted coordinates of the target [30].

## 4. Feasibility Verification of Giant Salamander Positioning Algorithm and System

### 4.1. Experimental Scheme Design

Initially, conduct underwater RSSI testing of the antenna to evaluate the feasibility of detecting RFID tags submerged in water. Subsequently, establish a three-dimensional spatial coordinate system in the field. By measuring the RSSI values of tags positioned 1 m from the antenna and the corresponding distances between them, calculate the parameters A (signal strength when the antenna and tag are 1 m apart) and *n* (environmental attenuation factor) in the RSSI-to-distance conversion formula. This process will facilitate the derivation of the log-distance path loss model. In the field, after collecting partial data, employ the GAIN model to generate additional data and validate the accuracy of the generated information. Input the generated data into various models to ascertain the final positions and compare the positioning results of different models to demonstrate the superiority of D-Tr positioning. Finally, evaluate the stability of the positioning system under varying conditions. (This study has been approved by the Institutional Ethics Committee (see Appendix A for details)).

### 4.2. Underwater RSSI Testing of Antenna

To evaluate the capability of the antenna in accurately identifying the underwater reference tag, we conducted tests under submerged conditions, as illustrated in Figure 6.

The testing process was specifically divided into the following three scenarios:

(1) When the antenna is positioned 0.20 to 0.35 m above the water surface, and the tag is submerged between 0.2 and 0.3 m underwater, it can be detected with a Received Signal Strength Indicator (RSSI) value of approximately −68 dBm, resulting in a recognition count of 1 to 2 times.

(2) When the antenna floats on the water surface while the tag remains submerged at a depth of 0.2 to 0.3 m, detection is feasible with a Received Signal Strength Indicator (RSSI) value of approximately −68 dBm, resulting in a recognition count of one to two instances.

(3) When the antenna is floating on the water surface, and the tag is submerged to a depth of less than 0.15 m underwater, detection occurs with an RSSI value of approximately −63 dBm, and the recognition count continues to increase.

From the aforementioned tests, it can be concluded that when the tag is submerged to a depth of 0.3 m in water, the antenna can successfully identify the tag.

This experiment involved positioning observations of the Chinese giant salamander during the dry season in the Zhongjian River, located in Xianfeng, Hubei. During this period, the river’s water depth was relatively shallow, typically remaining below 0.2 m (see Figure 7).

Based on the experimental results presented, it can be inferred that utilizing antennas to detect signal strength from tags attached to giant salamanders in their natural habitat, along with the application of the 3D LANDMARC positioning algorithm, is both theoretically valid and practically applicable for field positioning of giant salamanders.

### 4.3. Establishment of a Three-Dimensional Spatial Coordinate System in the Wild

Two antennas should be installed on each riverbank and in the river at the designated field site, referred to as Antenna D, Antenna E, Antenna F, and Antenna G, respectively. Additionally, a camera should be installed at each antenna location. The control box should be positioned midway between the two antennas on the riverbank. The coordinates of the field plane are illustrated in Figure 8.

### 4.4. Solution to Environmental Attenuation Factor

To obtain the log-distance path loss model, one must first measure *A*, which represents the signal strength when the antenna is positioned 1 m away from the tag. Subsequently, the RSSI values measured at various coordinates can be substituted to determine the environmental attenuation factor, *n*.

We installed four antennas above two breeding ponds where Chinese giant salamanders frequently engage in activities, positioning the control box centrally. The antennas were designated as Antenna A (−1.2, 0, 2.4), Antenna B (0, 0.9, 2.3), and Antenna C (0, −1.1, 2.4), each suspended at a height of 2.4 m above the ground. The experimental setup is illustrated in Figure 9 below.

The signal strength when the antenna is 1 m away from the tag is shown in Table 3.

By averaging the signal strengths from each antenna presented in Table 3, with the tag positioned at 1 m intervals, parameter *A* was computed to be −54.42 dBm. Five measurement points were strategically selected to maximize their distance from one another while still remaining detectable by the antennas: Point 1 (−0.4, −0.6, 0.3), Point 2 (−0.7, 0.15, 0.3), Point 3 (1.6, 0.6, 0.25), Point 4 (1, −0.3, 0.25), and Point 5 (1, 0.3, 0.3). Their precise locations are depicted in Figure 10. The measured distances to each antenna, along with the corresponding Received Signal Strength Indicator (RSSI) values, are detailed in Table 4.

The formula for converting RSSI to distance is as follows:(4)$$D_{ih}\;=\;10^{\frac{\text{A-RSSI}}{10n}}$$

In this context, *D_ih_* denotes the distance between the target tag *i* and antenna *h*, measured in meters, while *RSSI* indicates the signal strength of the target tag as detected by the antenna, expressed in dBm.

By inserting each observed value into Equation (8), the environmental attenuation factor *n* is determined to be 1.218. Subsequently, using the derived values of *A* and *n* in Equation (1) allows for the derivation of the log-distance path loss model.

### 4.5. Data Generation by GAIN 

#### 4.5.1. Raw Data Filtering

The precision of data generation is influenced by the RSSI readings from reference tags. In real-world scenarios, environmental factors or multipath phenomena can lead to significant variations in these RSSI readings, sometimes resulting in outlier values. Such outliers may closely resemble the RSSI values of the target tag, which can cause the position coordinates of these anomalous reference tags to be assigned an excessively high weight. In severe instances, the weight of an erroneous position coordinate could reach 1, leading to substantial inaccuracies in the positioning outcomes. Thus, ensuring consistent RSSI values from reference tags is essential for enhancing accuracy. The following are the specific methods for processing these data:

To begin with, a 3D LANDMARC positioning system should be set up, and the *n* groups of RSSI_i_ readings from the target tag or reference tags measured by antenna *h* need to be filtered. Collect 10 RSSI_i_ readings to form a dataset and then discard the highest and lowest values to minimize their influence on the average. Next, due to signal obstruction by physical barriers during transmission, which reduces the signal strength and consequently the RSSI readings, a higher threshold is applied to the lower values. Specifically, any sample data falling outside the range of (*u* − 0.5*σ*, *u* + *σ*) should be excluded (where *u* is the sample mean and *σ* is the sample standard deviation) to create a fingerprint database with more consistent RSSI values. Furthermore, if this approach does not adequately remove RSSI values that are too close to the target tag, leading to a correlation of zero with other reference tags, the correlation can be utilized for correction, and its value is defined as follows:(5)$$\partial =\frac{2}{3}E_{m}(E_{m}\;\neq \;0)$$

*E_m_* denotes the least non-zero correlation value found among *p* (where *p* > 0) reference tags chosen in proximity to the target tag.

By employing the previously described approach, we ultimately identified 600 datasets associated with Antenna D and generated an additional 10,000 datasets using the GAIN (generative adversarial imputation network) framework in conjunction with the log-distance path loss model.

#### 4.5.2. GAIN Model Configuration and Hyperparameters

This research employs an enhanced generative adversarial imputation network (GAIN) to facilitate the generation of RFID data. The generator processes inputs that consist of two-dimensional target location coordinates, 128-dimensional Gaussian noise, and two-dimensional outputs from the log-distance path loss model. It generates sequences of Received Signal Strength Indicator (RSSI) and RSSI differentials of a predetermined length using a five-layer fully connected architecture with layer sizes of 256, 512, 1024, and 512. Concurrently, the discriminator is structured as a four-layer one-dimensional convolutional neural network (CNN) with a kernel size of 5 and channel counts of 64, 128, 256, and 512, followed by a three-layer fully connected network, which is tasked with distinguishing between authentic and synthetic data. Key hyperparameters include a learning rate set at 2 × 10^−4^, a batch size of 64, and a total of 2000 training epochs, ensuring that the generated sequences comply with the principles of wireless propagation and accurately reflect the statistical properties of actual data.

After thorough screening, we ultimately selected 600 sets of data from Antenna D. Utilizing the GAIN data, we constructed a generative adversarial network (GAN) that generated 10,000 sets of data based on the logarithmic distance path loss model.

#### 4.5.3. Verification of the Availability of Generated Data

A total of 9000 data entries were utilized for training the model, while the remaining 1000 entries were reserved for testing its robustness. The model’s performance was evaluated using three key metrics: mean absolute error (MAE), root mean square error (RMSE), and the coefficient of determination (R^2^).

The data accuracy comparison chart (Figure 11) clearly illustrates that the MAE, RMSE, and R^2^ metrics for the adversarially generated data are similar to those of the filtered data. In contrast to the original dataset, the adversarially generated data shows a 22.08% reduction in MAE, a 14.59% reduction in RMSE, and a 43.55% increase in R^2^, indicating that the adversarially generated data is more accurate than the original.

### 4.6. Experimental Setup

The challenge of positioning giant salamanders using RFID technology can be framed as a regression problem, with the D-Tr model demonstrating effective results. To enhance the reproducibility of the experiments, it is crucial to explicitly outline all hyperparameters associated with the D-Tr model. The detailed configurations of these parameters are presented in Table 5 and Table 6.

### 4.7. Analysis of Ablation Experiment Results

To evaluate the performance of the GAIN generative adversarial network and the D-Tr model in localization tasks, we conducted ablation experiments utilizing the same dataset and maintaining consistent experimental settings. The results were compared with those obtained from the conventional 3D LANDMARC approach, and the findings are summarized in Table 7.

The data presented in Table 7 demonstrate that the integration of the GAIN with the D-Tr model resulted in significant reductions of 14.14%, 66.67%, 24.60%, and 59.26% in mean error, standard deviation, maximum error, and minimum error, respectively, compared to the D-Tr model evaluated in isolation. In contrast, when this combined method was assessed against the 3D LANDMARC approach, the reductions in the same error metrics were even more substantial, at 78.04%, 80.26%, 83.27%, and 94.53%. Furthermore, when the results were compared to those obtained using only the 3D LANDMARC method, the reductions in errors were recorded at 81.32%, 82.95%, 84.69%, and 94.79%.

The enhancement in performance is primarily attributed to two factors: the GAIN’s capacity to refine and improve the quality of raw data, and the D-Tr model’s proficient feature extraction capabilities that enable precise detection and application of distinguishing features in RSSI signals. Collectively, these elements not only enhance the overall accuracy of the positioning model but also bolster its resilience and adaptability in complex environments, significantly improving the effectiveness of positioning giant salamanders in challenging natural settings.

### 4.8. Comparison of Simulated Positioning Experiments of Various Models in the Wild

Due to the straightforward nature of the surroundings around antenna D, data collection is facilitated. This study utilizes antenna D as a case study in a natural setting to establish a coordinate framework, with the antenna positioned at (0, 0, 3), as illustrated in Figure 9. To simulate the movement of giant salamanders, RFID tags are progressively relocated from close proximity to a distance, capturing a dataset that includes 10 coordinates along with their respective RSSI values. These coordinates are (0.5, 0.6, 3.0), (0.3, 1.3, 1.0), (0.6, 0.5, 0.4), (0.3, 1.8, 0.2), (1.2, 1.2, 2.0), (0.9, 1.5, 1.5), (1.8, 1.8, 0.5), (2.1, 2.4, 2.7), (3.0, 2.1, 2.1), and (2.1, 1.8, 2.0). Following the data filtration process, additional data are generated using the GAIN, and positioning is executed with the previously trained technique. The results of the positioning are presented in Figure 12 and Table 8.

The analysis of the three-dimensional coordinate axes illustrated in Figure 12 indicates that the D-Tr model surpasses its counterparts in positioning accuracy. As detailed in Table 8, the average error for the D-Tr model is measured at 6.118% along the X-axis, 3.310% on the Y-axis, and 2.143% for the Z-axis. In comparison to other models, the D-Tr model demonstrates improvements of 20.075%, 15.331%, and 12.925% on the X, Y, and Z axes, respectively. These results highlight the D-Tr model’s exceptional reliability and superiority in positioning capabilities.

#### Statistical Significance Analysis

In order to assess if the D-Tr model provides superior positioning accuracy in coordinate localization relative to alternative models, a *t*-test can be utilized for result analysis. It is assumed that the discrepancies between the coordinates identified by the D-Tr model and those from other models adhere to a normal distribution. The hypotheses for the two-tailed *t*-test are outlined as follows [31,32]:

Null hypothesis: The mean errors of both models are zero.

Alternative hypothesis: The mean errors of the two models are significantly different from 0.

The significance level, denoted as α, serves as a benchmark for assessing the statistical significance of results, commonly established at α = 0.05.

Utilizing the three-dimensional positional coordinates, determine the Euclidean distance error for every sample point by applying the following formula:(6)$${Error}_{i}=\sqrt{{\left(x_{i}-{\hat{x}}_{i}\right)}^{2}+{\left(y_{i}-{\hat{y}}_{i}\right)}^{2}+{\left(z_{i}-{\hat{z}}_{i}\right)}^{2}}$$

Here, (*x_i_*, *y_i_*, *z_i_*) represents the sample coordinates, (${\hat{x}}_{i},\;{\hat{y}}_{i},\;{\hat{z}}_{i}$) represents the predicted coordinates.

The formula for calculating the statistic is as follows:(7)$$s=\sqrt{\frac{\sum_{i=1}^{n}{\left(x_{i}-\bar{x}\right)}^{2}}{n}}$$

In this context, *s* denotes the variance of the error measurements for the tags being evaluated, m; *x_i_* indicates the error measurement for the *i*-th tag under evaluation, m; $\bar{x}$ represents the error measurements for the tags being evaluated, m; and *n* signifies the total count of error measurements for the tags under evaluation.

Degrees of freedom are determined as follows:(8)df=n−1

In this context, df denotes the quantity of limitations placed on the sample data (the labeling inaccuracies to be assessed) during the estimation of population parameters.

The discrepancy is determined by evaluating Formulas (6) and (7) against the sample coordinates, as illustrated in the following Table 9.

The *t*-test for a single sample, available in SPSS statistical software (version 29) (α = 0.05), was employed to conduct *t*-tests across all possible pairwise combinations of the five data groups presented in Table 9. The results are displayed in Table 10 below.

In Table 10, the *t*-value serves as a metric in the *t*-test that assesses the extent of variation between the sample statistic and the population parameter. When the absolute *t*-value exceeds the critical threshold, it leads to the rejection of the null hypothesis, signifying a significant difference between the sample data and the null hypothesis. Conversely, if the absolute *t*-value is less than or equal to the critical threshold, the null hypothesis remains unchallenged. The *p*-value indicates the probability of obtaining the observed sample data or more extreme outcomes, assuming the null hypothesis is true. A lower *p*-value suggests a more pronounced difference between the sample data and the null hypothesis, thereby providing stronger justification for rejecting the null hypothesis. In contrast, a higher *p*-value implies insufficient evidence to dismiss the null hypothesis. Effect size is a measure that evaluates the strength of the treatment effect. A negative effect size indicates that the error of the first model is less than that of the second model, while a positive effect size suggests the opposite. The absolute value of the effect size reflects the strength of the effect.

Table 10 illustrates that, in contrast to other models, the D-Tr model exhibits *p*-values from correlation tests that fall below 0.05, along with *t*-statistics and effect sizes also below 0. This evidence supports the rejection of the null hypothesis, indicating that the D-Tr model’s positioning accuracy significantly surpasses that of its counterparts.

### 4.9. Verification of the Stability of the D-Tr Model’s Positioning System

The D-Tr model exhibits superior accuracy in positioning systems compared to alternative models; however, its reliability requires further validation. This study evaluates its positioning precision across various scenarios, with detailed results presented in Table 11 below.

The data presented in Table 11 clearly indicate that the mean discrepancies between the positioning outcomes and the true locations across various conditions—sunny, cloudy, rainy, and nighttime—are 1.445%, 1.399%, 4.619%, and 0.762%, respectively. The D-Tr model maintains a positioning accuracy within a 5% error margin across these different environments. This performance demonstrates that the model adequately meets the accuracy standards required for various scenarios in the Zhongjian River National Giant Salamander Nature Reserve, located in Xianfeng, Hubei Province, and exhibits strong reliability.

### 4.10. System Application

This project management platform provides a direct visualization of positioning outcomes, facilitating the tracking and positioning of experimental giant salamanders. This capability enhances the examination of their behavioral patterns and movement paths, resulting in improved insights into the lifestyles of wild giant salamanders. Figure 13 below illustrates the interface of the management platform, where red dots indicate the tags of the giant salamanders under study, blue dots denote reference tags, and black dots represent the antennas.

Upon detecting the Received Signal Strength Indicator (RSSI) signal, the camera is activated to capture images simultaneously. These images are subsequently utilized as the dataset for training the image recognition model. This trained model is then implemented on the host system. Once the model is prepared, it is capable of recognizing wild giant salamanders within the camera’s field of view, facilitating the automatic capture and storage of photos and videos, as illustrated in Figure 14 below.

## 5. Conclusions

This research tackles the issue of closely monitoring the behavior of wild Chinese giant salamanders in intricate aquatic settings by introducing and validating a passive ultra-high-frequency radio frequency identification (UHF RFID) positioning system utilizing a dual-stage transformer (D-Tr). Initially, the system employs a generative adversarial network (GAIN) alongside the log-distance path loss model to effectively reconstruct lost received signal strength indication (RSSI) data that environmental barriers may obscure. The reconstructed data shows a 22.03% improvement in mean absolute error (MAE) and a 27.60% reduction in root mean square error (RMSE) when compared to the original data, greatly enhancing data reliability. Furthermore, the D-Tr model has been creatively applied to animal localization. By simultaneously analyzing spatiotemporal features from both the original RSSI data and their first-order differentials, the system achieved a localization precision of ≤10 cm in field tests. In comparison to conventional sequential models like LSTM and GRU, it exhibited average enhancements of 20.075%, 15.331%, and 12.925% along the X, Y, and Z axes, respectively. Additionally, it maintained consistent performance (error < 5%) across diverse weather and lighting conditions, showcasing remarkable robustness. This centimeter-level localization ability facilitated quantitative assessments of the cave-dwelling behaviors of giant salamanders, including activity range and duration of stay. The “data generation–feature fusion–precise localization” technological framework developed by this research institute not only offers reusable monitoring solutions for endangered species reliant on caves and water bodies but also advances the shift in species conservation from passive oversight to proactive, intelligent management driven by spatiotemporal big data through the incorporation of visual triggering mechanisms.

## 6. Outlook

This study validated the feasibility and accuracy of the system within the mountain stream environment of the Hubei Xianfeng Zhongjian River Chinese Giant Salamander National Nature Reserve. However, the habitats of Chinese giant salamanders in China are diverse, encompassing environments with varying hydrological and geological characteristics, such as plateau streams, karst cave rivers, and lowland plain rivers. Furthermore, while the attachment of RFID tags to the tails of giant salamanders facilitates continuous, non-invasive positioning and monitoring, it may still impact their natural behaviors. During this study, the giant salamanders in the reserve entered a state of seasonal dormancy, which prevented simultaneous behavioral response observation experiments following tag attachment. In future phases, controlled behavioral experiments will be conducted to systematically assess the potential effects of tag attachment on individual activity rhythms, den utilization, foraging behavior, and more, with the goal of further optimizing tagging protocols and enhancing the ecological authenticity of monitoring data. To improve the system’s universality and ecological applicability, multi-environment validation tests should be conducted in the aforementioned typical habitats to evaluate the system’s positioning stability and accuracy under varying conditions of water flow velocity, water turbidity, substrate composition, and riparian vegetation. Through cross-regional and multi-habitat comparative studies, model parameters and hardware deployment strategies can be further refined to develop a standardized and scalable positioning and monitoring solution applicable to various giant salamander habitats. This will provide a reliable technical framework for the conservation of endangered aquatic species in similar habitats, both nationally and globally.

## Figures and Tables

**Figure 1 sensors-26-00106-f001:**
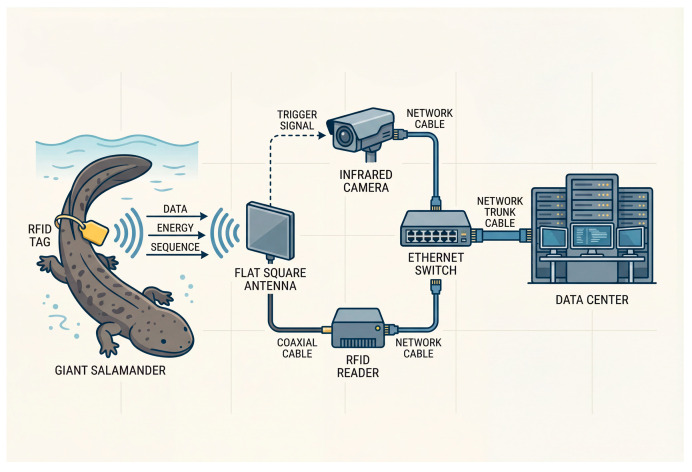
RFID: giant salamander positioning process.

**Figure 2 sensors-26-00106-f002:**
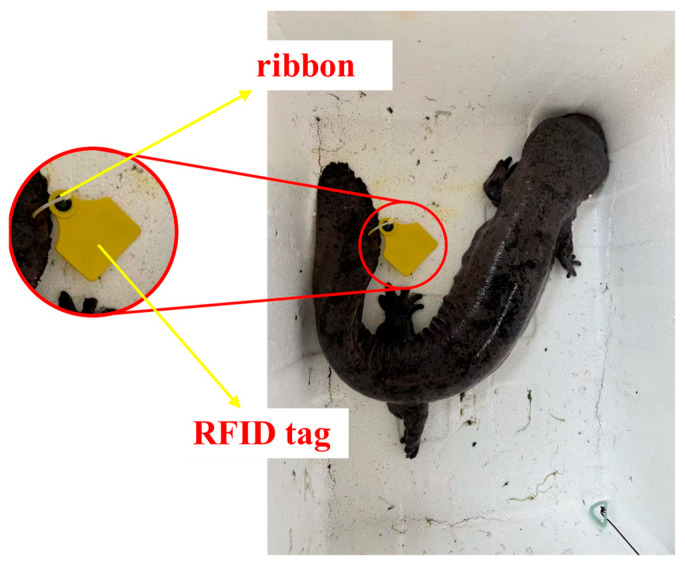
Giant salamanders wearing UHF RFID tags.

**Figure 3 sensors-26-00106-f003:**
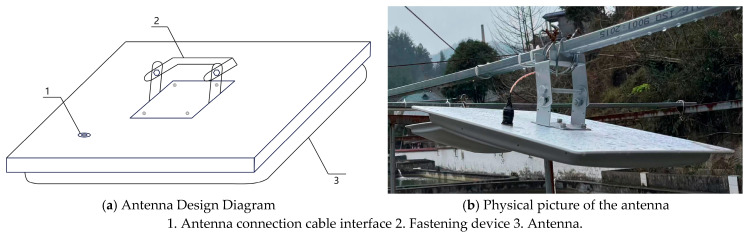
Comparison diagram between the antenna design and the actual antenna.

**Figure 4 sensors-26-00106-f004:**
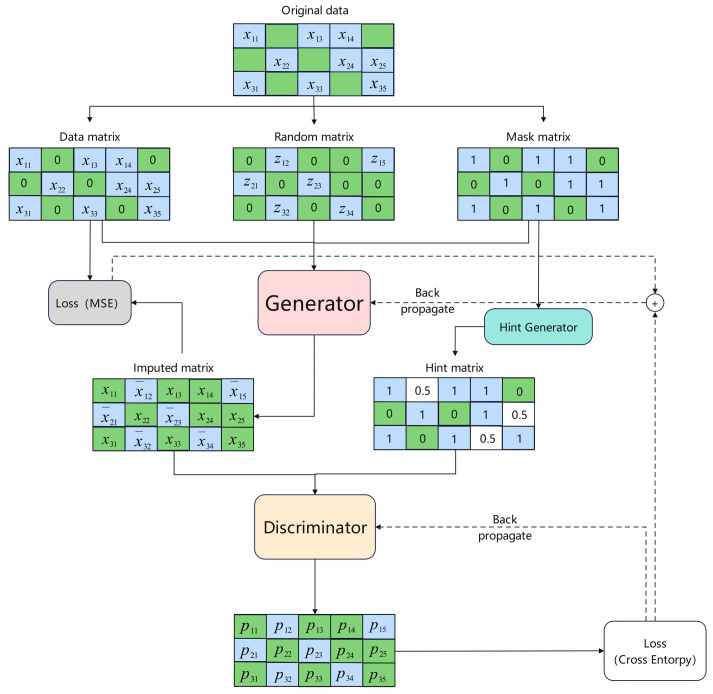
Structural framework of the GAIN.

**Figure 5 sensors-26-00106-f005:**
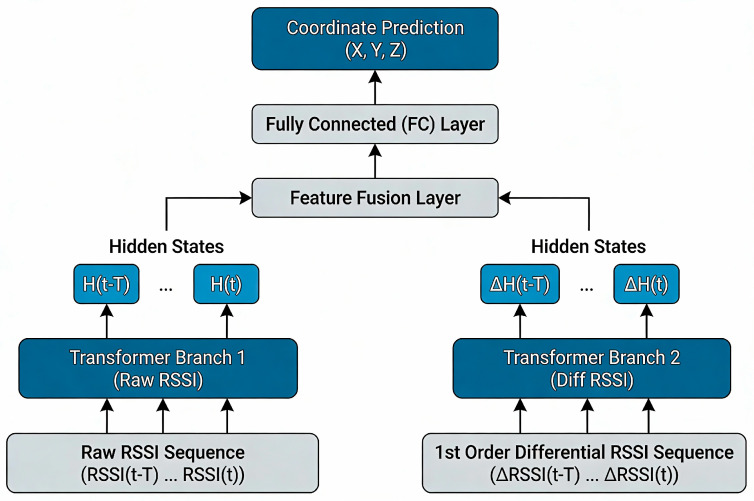
Model structure of a two-stage transformer. The ellipsis (…) is used as a compact notation to represent the complete sequence of values between the given start and end points.

**Figure 6 sensors-26-00106-f006:**
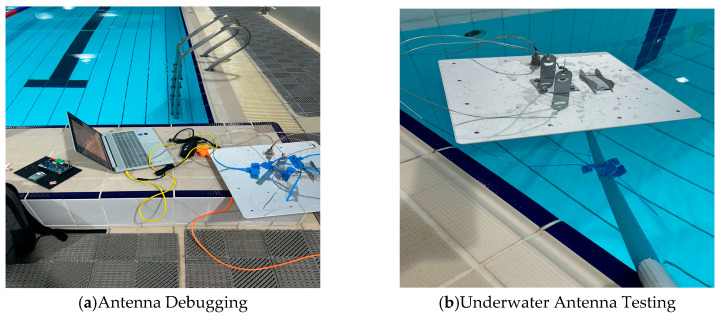
Underwater test. (**a**) Antenna Debugging; (**b**) Underwater Antenna Testing.

**Figure 7 sensors-26-00106-f007:**
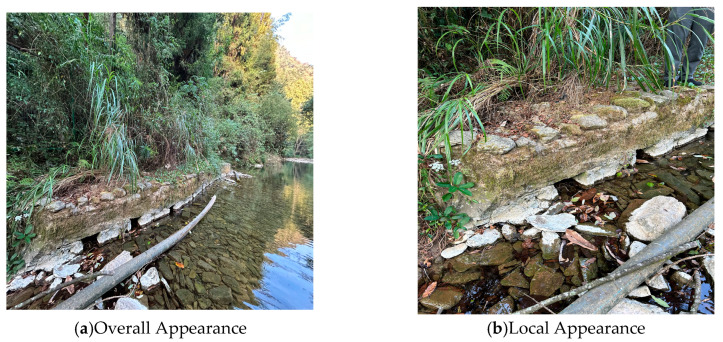
Dry season of Zhongjian River in Xianfeng County, Hubei Province. (**a**) Overall Appearance; (**b**) Local Appearance.

**Figure 8 sensors-26-00106-f008:**
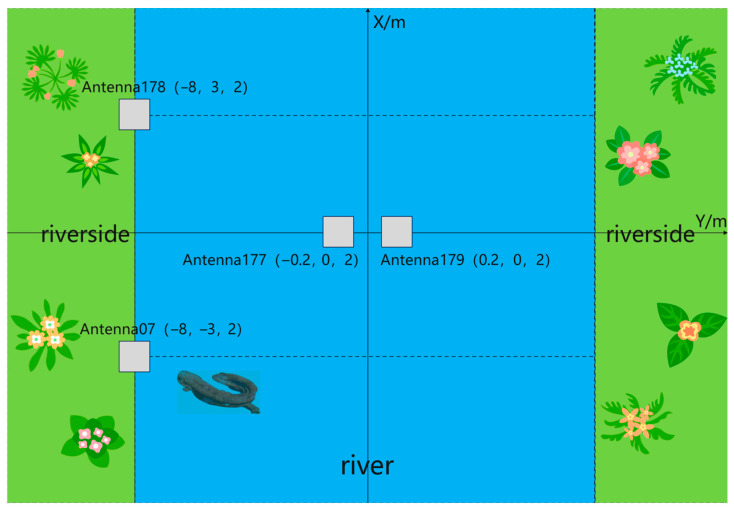
Field plane coordinate diagram.

**Figure 9 sensors-26-00106-f009:**
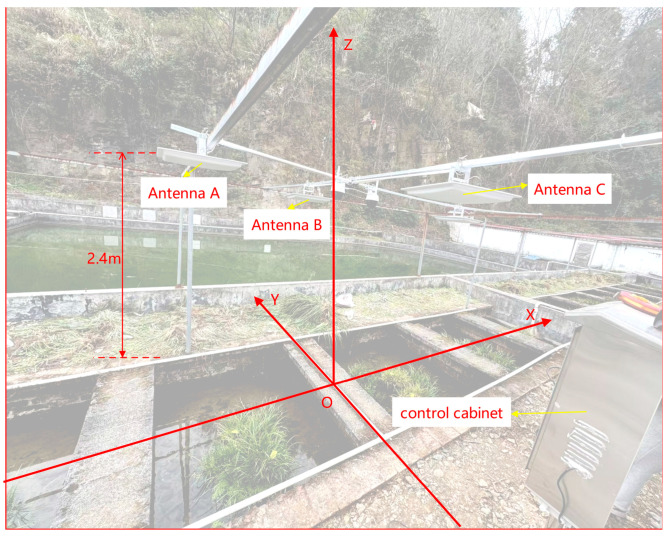
Experimental setup diagram.

**Figure 10 sensors-26-00106-f010:**
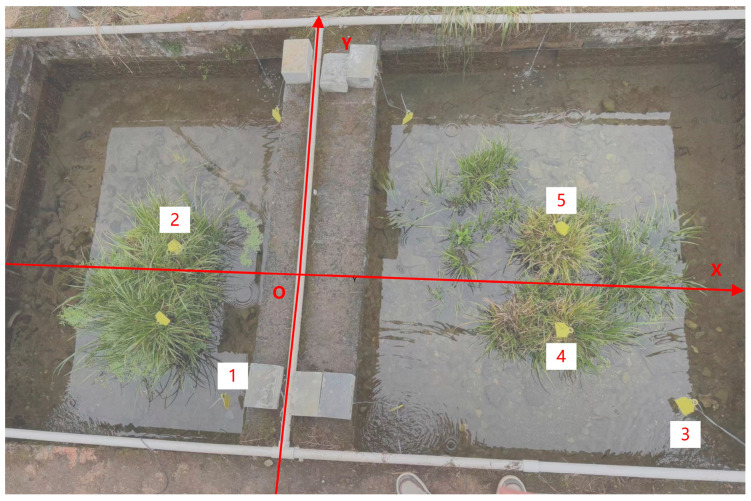
Planar coordinate diagram. The numbers in the figure are a visual representation of the actual positions of the five coordinates listed in Table 4.

**Figure 11 sensors-26-00106-f011:**
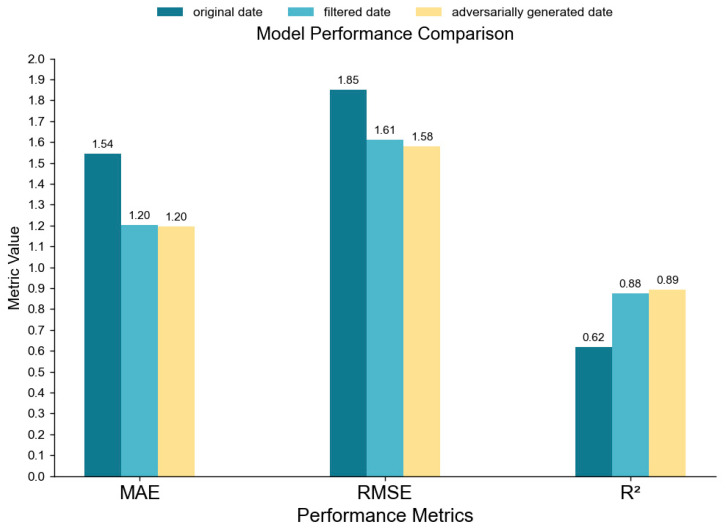
Accuracy comparison chart of various data types.

**Figure 12 sensors-26-00106-f012:**
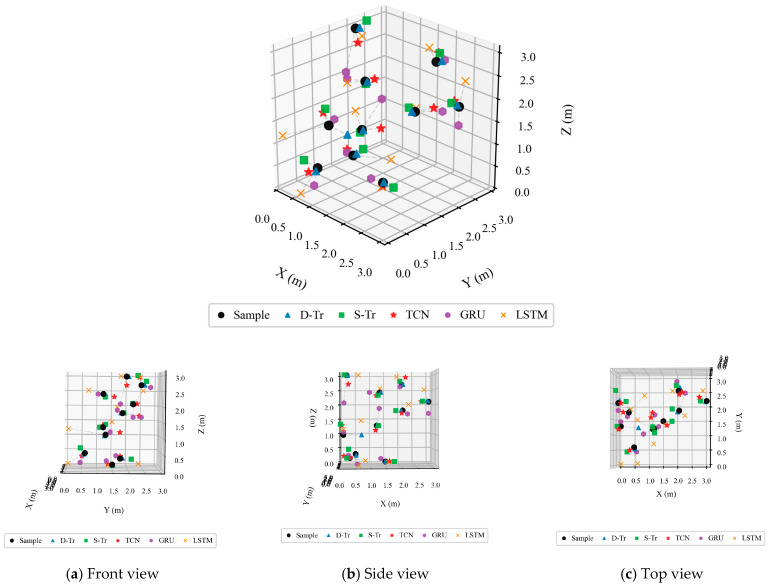
Positioning results. (**a**) Front view; (**b**) Side view; (**c**) Top view.

**Figure 13 sensors-26-00106-f013:**
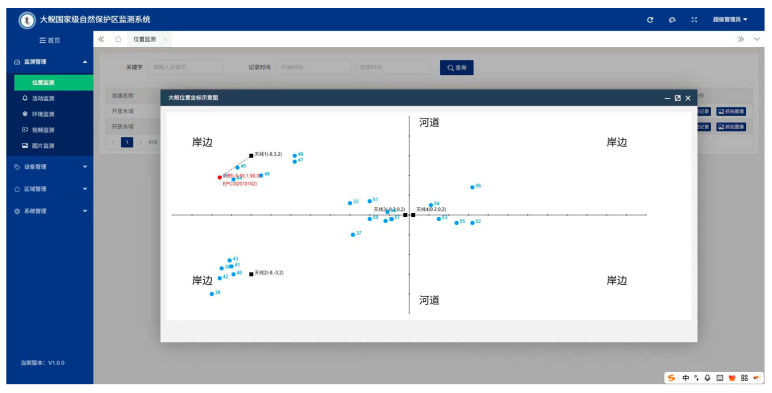
Positioning interface. The Chinese label “天线” means “Antenna”. The Chinese label “大鲵” means “Giant Salamander”. The Chinese label “河道” means “River Channel”. The Chinese label “岸边” means “Bank”.

**Figure 14 sensors-26-00106-f014:**
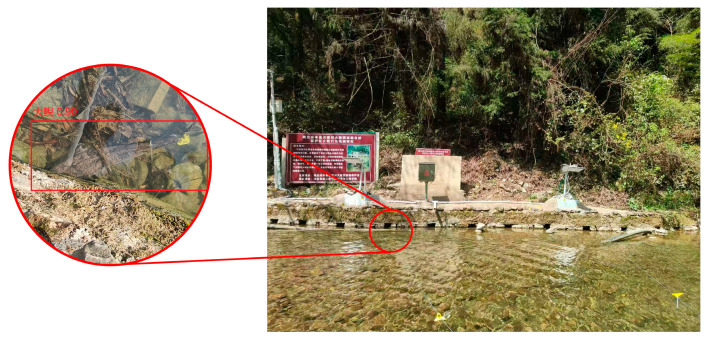
Giant salamander image recognition results. The Chinese label “大鲵” means “Giant Salamander”.

**Table 1 sensors-26-00106-t001:** Antenna-related parameters.

Parameter Name	Value
Operating Frequency	902–928 MHz
Operating Protocol	ISO18000-6C/GEN 2
Built-in Antenna	12 db
Communication Interface	USB/232/485/WiFi/TCP/Relay
Radio Frequency Chip	Impinj E710
Output Power	0–33 dBm
Power Consumption	≤10 W
Operating Voltage	DC 9–12 V

**Table 2 sensors-26-00106-t002:** Reader-related parameters.

Parameter Name	Value
Operating Frequency	902–928 MHz
Operating Protocol	ISO18000-6C/T29768/GJB7377.1
Communication Interface	USB/RS232/485/Ethernet Port/IO Relay
Tag Cache	1000 Tags

**Table 3 sensors-26-00106-t003:** Signal intensity table when the antenna is 1 m apart from the label.

Antenna A RSSI Value/dBm	Antenna B RSSI Value/dBm	Antenna C RSSI Value/dBm
−58.00	−54.00	−51.00
−58.00	−53.00	−49.00
−62.00	−56.00	−47.00
−54.00	−56.00	−47.00
−62.00	−53.00	−50.00
−65.00	−51.00	−50.00
−64.00	−52.00	−49.00
−62.00	−52.00	−50.00
−63.00	−51.00	−48.00
−64.00	−51.00	−49.00
−65.00	−48.00	−47.00
−64.00	−54.00	−50.00

**Table 4 sensors-26-00106-t004:** The RSSI value and distance from the coordinates to each antenna.

	Antenna	Antenna A		Antenna B		Antenna C	
Coordinate		RSSI/dBm	Distance/m	RSSI/dBm	Distance/m	RSSI/dBm	Distance/m
Coordinate 1		−42.83	2.33	−58.67	2.53	−60.5	2.2
Coordinate 2		−54.33	2.16	−57.17	2.25	−60.33	2.54
Coordinate 3		−69	3.58	--	--	−67	2.73
Coordinate 4		--	--	--	--	−62.17	2.5
Coordinate 5		--	--	--	--	−66.83	2.75

**Table 5 sensors-26-00106-t005:** Model architecture parameters.

Parameter Name	Parameter
number of model layers	9
embedding dimension	128
activation function	GELU
number of attention heads	8

**Table 6 sensors-26-00106-t006:** Training parameters.

Parameter Name	Parameter
learning rate	0.001
Batch size	32
training rounds	1000
optimizer	AdamW

**Table 7 sensors-26-00106-t007:** Results of ablation experiments.

3D LANDMARC	GAIN	D-Tr	Average Error/m	Standard Deviation/m	Maximum Error/m	Minimum Error/m
√	×	×	0.455	0.264	0.921	0.422
√	√	×	0.387	0.228	0.843	0.402
×	×	√	0.099	0.135	0.187	0.054
×	√	√	0.085	0.045	0.141	0.022

Note: √ indicates adopting this method, × indicates not adopting this method.

**Table 8 sensors-26-00106-t008:** 3D positioning coordinates of each model.

Name	Sample Coordinates	D-Tr	S-Tr	TCN	GRU	LSTM
coordinates/m	(0.20, 2.00, 3.00)	(0.24, 2.08, 3.00)	(0.12, 2.40, 3.04)	(0.26, 2.01, 2.70)	(0.11, 1.80, 2.06)	(0.60, 1.80, 3.00)
	(0.10, 1.30, 1.00)	(0.12, 1.26, 1.00)	(0.00, 1.30, 1.35)	(0.05, 1.20, 1.31)	(0.10, 1.46, 1.09)	(0.11, 0.03, 1.20)
	(0.50, 0.60, 0.40)	(0.52, 0.54, 0.36)	(0.27, 0.44, 0.56)	(0.34, 0.50, 0.29)	(0.56, 0.44, 0.08)	(0.60, 0.04, 0.06)
	(0.30, 1.80, 0.20)	(0.32, 1.88, 0.22)	(0.20, 2.19, 0.18)	(0.11, 1.81, 0.27)	(0.26, 1.66, 0.30)	(0.84, 2.40, 0.08)
	(1.20, 1.20, 1.30)	(1.22, 1.21, 1.30)	(1.26, 1.09, 1.30)	(1.18, 1.76, 1.14)	(1.30, 1.66, 1.87)	(0.69, 1.51, 1.46)
	(1.30, 1.20, 2.40)	(1.34, 1.22, 2.40)	(1.25, 1.26, 2.31)	(1.20, 1.56, 2.31)	(0.96, 1.03, 2.40)	(1.30, 0.71, 2.52)
	(1.50, 1.50, 0.10)	(1.50, 1.52, 0.10)	(1.80, 1.50, 0.10)	(1.63, 1.36, 0.12)	(1.34, 1.31, 0.21)	(1.59, 1.36, 0.09)
	(2.10, 2.40, 2.70)	(2.15, 2.51, 2.70)	(2.04, 2.56, 2.82)	(2.23, 2.33, 2.95)	(2.05, 2.70, 2.63)	(1.92, 2.38, 2.96)
	(3.00, 2.10, 2.10)	(2.97, 2.08, 2.13)	(2.80, 2.10, 2.12)	(2.76, 2.23, 2.11)	(3.00, 2.10, 1.70)	(2.88, 2.41, 2.52)
	(2.10, 1.80, 1.80)	(2.05, 1.76, 1.80)	(1.87, 1.86, 1.80)	(2.10, 2.34, 1.71)	(2.30, 2.39, 1.68)	(2.29, 1.63, 2.00)

Note: The coordinates in the table are spatial coordinates (x, y, z).

**Table 9 sensors-26-00106-t009:** Statistics of localization errors of each model.

Model Name	Mean Error ± Standard Deviation/m	Median Error/ m	Minimum Error/m	Maximum Error/m	RMSE/m
D-Tr	0.085 ± 0.045	0.080	0.022	0.141	0.096
S-Tr	0.312 ± 0.215	0.255	0.112	0.724	0.380
TCN	0.210 ± 0.125	0.185	0.064	0.438	0.243
GRU	0.386 ± 0.298	0.305	0.091	0.906	0.487
LSTM	0.335 ± 0.281	0.268	0.090	0.972	0.439

Please be aware that RMSE, which represents root mean square error, is derived from the calculations performed on ten sample data points.

**Table 10 sensors-26-00106-t010:** Paired Comparison Significance Test of Model Performance.

Comparative Pair	*t*-Statistic	*p*-Value	Whether Significant (α = 0.05)	Effect Size
D-Tr vs. S-Tr	−3.245	0.010	yes	−1.450
D-Tr vs. TCN	−3.521	0.006	yes	−1.570
D-Tr vs. GRU	−3.128	0.012	yes	−1.400
D-Tr vs. LSTM	−2.876	0.018	yes	−1.290
S-Tr vs. TCN	1.234	0.248	no	0.550
S-Tr vs. GRU	−0.845	0.420	no	−0.380
S-Tr vs. LSTM	−0.321	0.755	no	−0.140
TCN vs. GRU	−1.765	0.112	no	−0.790
TCN vs. LSTM	−1.432	0.186	no	−0.640
GRU vs. LSTM	0.543	0.600	no	0.240

**Table 11 sensors-26-00106-t011:** The positioning effect of the D-Tr model in different scenarios.

Scenario	Actual Position	Sunny Environment	Cloudy Environment	Rainy Environment	Night Environment
coordinates/m	(1.20, 2.20, 1.00)	(1.21, 2.22, 0.98)	(1.21, 2.23, 1.03)	(1.24, 2.16, 0.95)	(1.22, 2.20, 1.01)
	(0.30, 0.20, 1.20)	(0.30, 0.20, 1.18)	(0.30, 0.19, 1.21)	(0.28, 0.22, 1.26)	(0.30, 0.20, 1.19)
	(2.10, 1.40, 2.30)	(2.08, 1.41, 2.32)	(2.10, 1.41, 2.31)	(2.08, 1.35, 2.24)	(2.11, 1.41, 2.31)
	(1.50, 1.60, 0.80)	(1.49, 1.58, 0.79)	(1.48, 1.61, 0.80)	(1.44, 1.64, 0.76)	(1.49, 1.60, 0.80)
	(0.90, 1.20, 1.60)	(0.91, 1.21, 1.62)	(0.91, 1.21, 1.58)	(0.94, 1.22, 1.56)	(0.90, 1.21, 1.61)
	(0.90, 2.80, 2.50)	(0.90, 2.82, 2.51)	(0.91, 2.81, 2.52)	(0.88, 2.73, 2.57)	(0.91, 2.82, 2.49)
	(1.20, 1.60, 1.80)	(1.20, 1.61, 1.82)	(1.20, 1.62, 1.81)	(1.11, 1.54, 1.74)	(1.20, 1.61, 1.79)

Note: The coordinates in the table are spatial coordinates (x, y, z).

## Data Availability

The data supporting the results reported in this study are not publicly available due to privacy restrictions.

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
