# Peer review of "Study on the Localization Technology for Giant Salamanders Using Passive UHF RFID and Incomplete D-Tr Measurement Data"

_sensors, 2025, doi:10.3390/s26010106_

Round 1

Reviewer 1 Report

Comments and Suggestions for Authors

The authors have presented a typical research manuscript focused on the study of localization algorithms for tracking wild giant salamanders in their natural habitat. The primary contribution of the paper lies in comparing the effectiveness of various positioning algorithms and introducing a novel approach based on a Dual-stage Transformer (D-Tr) framework.

Resent research presented in literature supports the claim that dual-stage or multi-stage Transformer models can significantly improve localization accuracy by better capturing the temporal dynamics of signal variations. In this work, the Authors benchmark their proposed D-Tr approach against several alternatives, including S-Tr, TCN, GRU, LSTM models.

The dataset analysed originates from a localization system designed for giant salamander in their natural environment, which employs passive UHF RFID technology. Since the collected RSSI signal strength is incomplete, the Authors incorporate ana enhanced 3D LANDMARC algorithm alongside a GAIN model to reconstruct missing RSSI values. This integration is a strong point of the study, as imputing missing data is critical for improving reliability in real-world deployments.

However, the most significant concern in this experiment is the use of an RFID system operating in the UHF band. As the Authors themselves acknowledge, RSSI is highly susceptible to environmental interference (metal, moisture, obstacles). This issue is only briefly mentioned in the manuscript, whereas it deserves more emphasis. Although antennas and tags might be protected against environmental conditions, RFID systems are generally not designed for underwater applications. Water strongly attenuates UHF radio waves due to its high dielectric constant, which leads to substantial absorption of electromagnetic energy. This limitation critically affects system performance in the salamander natural habitat, which includes aquatic environments. The Authors have to address this concern thoroughly and provide detailed explanations or mitigation strategies in the manuscript. This discussion is essential to ensure the practical applicability and robustness of the proposed implementation.

Additional remarks:

- At the beginning of the manuscript, the Authors provide a brief review of contemporary studies related to the topic; however, they completely omit in-text citations. Furthermore, they should use a standard referencing system used in technical scientific articles – specifically, a citation style where references are numbered according to the order they appear in the text.

- The text contains numerous editorial and punctuation errors, particularly in the descriptions of tables and figures. There are also issues such as unnecessary splitting of tables across pages and leaving isolated headings on a page preceding the chapter content.

- The resolution of the figures is unsatisfactory.

- Figure 1 is trivial, as it depicts a standard planar antenna commonly used in UHF RFID systems. Instead, the Authors should place greater emphasis on the challenges related to antenna placement and oriented within the target environment as a whole.

- The Authors should provide sufficient information to identify the RFID devices (antenna, reader) used. E.g., The IE704 reader is unfamiliar for Readers of the Journal, and details should be included to allow recognition of its manufacturer and specifications. Furthermore, since this is a commercial device nor designed by the Authors there is no need to include its photograph (Figure 2) in the manuscript. Its physical appearance has no significant impact on interaction with the target environment.

- The article lacks information regarding the strategy for tag placement in the tested environment, and this strategy remains completely unclear after analysing Figure 10.

- The formulas are presented carelessly: variables should be written in italics, and all variable symbols must be explicitly explained in the text. Consistency should be maintained in writing numerical indices. The units should be separated by a space from the preceding numeric value.

- There is a lack of consistency in the naming convention. It is unclear that designation such as Antenna 178, Antenna 07 … represents and how the relate to the labels used in Table 1 – Antenna A, Antenna B … and later in Table 2 – Antenna 1, Antenna 2…

- In Table 1: “Antenna” is not a physical quantity expressed in dBm. It is unclear which parameters changes across the rows of the tables.

- “The average signal strength, denoted as A, is determined to be -53.63 dBm” – the measurement data indicate a different value.

- It is unclear what the indictors ‘Coordinate 1-5’ refer to. The points should be appropriate mark on figures or defined in another way.

- Instead of including Table 3 and 4, it would better to improve the readability of the 3D plots. Additionally, abbreviations for neutral networks architectures have been introduced in the figures but not revealed. For completeness, these should be explained either in the text or in the figure caption.

Reviewer 2 Report

Comments and Suggestions for Authors

The paper aims to investigate a hybrid localization method for improving the positioning accuracy of the Chinese giant salamander. However, the following issues have been identified by the reviewer:

  1. A strong recommendation is made to improve the quality of all figures. Figure sizes should be uniform, and Figures 1 and 10 suffer from low resolution, inconsistent and excessively small font sizes, unclear content representation, and poor layout and color schemes. Layout, typography, and labeling should be enhanced for clarity and professionalism.
  2. The manuscript formatting should be revised: the main text should be justified, all figures and tables should be centered, with consistent spacing before and after. Additionally, equation fonts should match the body text size, and variables (e.g., *n*) must be set in italics to conform to academic standards.
  3. Technical terms such as "D-Tr model" are introduced without definition. All abbreviations should be spelled out upon first appearance, and terminology should be used consistently throughout the manuscript.
  4. Explanations provided below equations are overly verbose and written in a colloquial style. These should be rewritten concisely and formally, beginning directly with technical descriptions using academic language.
  5. The first and second paragraphs of the Introduction present overlapping content and should be merged into a single paragraph. The logical flow between sentences is weak, and the language is informal. For instance, phrases like “unfortunately” and “there” should be avoided.
  6. The literature review on domestic and international research status should cite relevant scholarly references rather than merely listing names. The discussion should be more concise and explicitly linked to the current study’s objectives and context.
  7. The main contributions of the paper should be clearly summarized and listed in bullet points at the end of the Introduction section.
  8. The Conclusion section should be more concise and presented as a single, cohesive paragraph. The additional contributions currently described are unclear and would be better organized in a dedicated table for improved readability.
  9. Attaching tags to the tail of the giant salamander may affect its natural behavior. It is recommended to conduct post-tagging behavioral observation experiments to assess potential impacts on locomotion or ecological performance.
  10. The experiments were conducted solely within the Hubei Xianfeng Zhongjianhe Giant Salamander National Nature Reserve. To evaluate generalizability, testing should be extended to other types of salamander habitats (such as highland streams, karst cave rivers, and lowland plain rivers) under diverse environmental conditions.
  11. The argumentation in the paper is limited and lacks sufficient comparative analysis. A more rigorous evaluation is required, comparing the proposed hybrid localization method against alternative approaches (e.g., traditional RSSI-based, machine learning-enhanced, or model-driven methods).

Reviewer 3 Report

Comments and Suggestions for Authors

The manuscript presents a technically interesting and practically relevant study on RFID-based localization for giant salamanders using a GAIN-assisted data reconstruction framework and a dual-stage Transformer (D-Tr) model. The integration of passive UHF RFID, deep learning-based time-series modeling, and real field deployment in a protected natural reserve represents a meaningful contribution to both sensor technology and ecological monitoring. The reported positioning accuracy and multi-scenario validation are encouraging.

However, several important issues should be addressed before the manuscript can be considered for publication:
• The architecture of the Dual-stage Transformer (D-Tr) is not described in sufficient detail. The authors should specify the number of layers, attention heads, embedding dimensions, activation functions, and training parameters (learning rate, batch size, number of epochs).
• The configuration and hyperparameters of the GAIN model are not clearly reported, which limits the reproducibility of the data generation process.
• The amount of real measured data is relatively small compared to the large volume of synthetically generated data. The risk of overfitting and the generalization capability of the model should be discussed more thoroughly, and additional real validation data would strengthen the study.
• The performance comparison with LSTM, GRU, TCN, and S-Tr lacks statistical significance analysis (e.g., confidence intervals or hypothesis testing).
• Several figures and tables require clearer captions, consistent units, and improved readability (font size, resolution, axis labels).
• The English language quality needs professional revision due to frequent grammatical errors and awkward phrasing.

Overall, the study is technically promising and practically relevant, but major revision is required before acceptance.

Comments on the Quality of English Language

The manuscript contains a noticeable number of grammatical errors, awkward expressions, and unclear phrasing. A thorough professional English language editing is strongly recommended to improve clarity and readability.

Round 2

Reviewer 1 Report

Comments and Suggestions for Authors

I would like to acknowledge the considerable effort the Authors have invested in revising the manuscript in response to the reviewers’ comments. The changes introduced are substantial and demonstrate a clear attempt to improve the quality of the work.

I recognize that the proposed solution presented in the manuscript is innovative and of potential interest to the journal’s readership. The work offers valuable insights and contributes meaningfully to the field.

However, please note that system of the in-text citation is still inadequate. Although the Authors may consider it improved, it has not been properly revised. This issue may have arisen during the conversion from text-editor to PDF format. This can be corrected during standard editorial processing.

Reviewer 2 Report

Comments and Suggestions for Authors

Accept in present form.

Reviewer 3 Report

Comments and Suggestions for Authors

The revised manuscript has been significantly improved. The authors have provided detailed descriptions of the D-Tr and GAIN architectures, including hyperparameters and training settings, and have added appropriate statistical significance analysis to support the comparative results. These revisions substantially enhance the reproducibility and scientific rigor of the study.

Minor issues remain regarding English language polishing and figure readability, which should be addressed to further improve clarity. Overall, the manuscript now represents a solid and well-supported contribution.

Comments on the Quality of English Language

The English language has been improved compared to the previous version. Minor grammatical issues and stylistic inconsistencies remain, and a final professional language polishing is recommended to further enhance clarity and readability.
